# Environmental and Economic Implication of Implementation Scale of Sewage Sludge Recycling Systems Considering Carbon Trading Price

**Jiawen Zhang [1],\*, Zhiyi Liang [1] , Toru Matsumoto [2] and Tiejia Zhang [1]**

[1]  Graduate School of Environmental Engineering, The University of Kitakyushu,
    Kitakyushu 808-0135, Fukuoka, Japan; liangzhiyi68@163.com (Z.L.); b0dac001@eng.kitakyu-u.ac.jp (T.Z.)
[2]  Institute of Environmental Science and Technology, The University of Kitakyushu,
    Kitakyushu 808-0135, Fukuoka, Japan; matsumoto-t@kitakyu-u.ac.jp
\*  Correspondence: tjzhangjiawen@163.com

**Abstract:** With China's ongoing economic development and increasing emphasis on environmental protection, the number and treatment capacity of sewage plants is increasing annually. Simultaneously, sludge production is increasing. In recent years, researchers have investigated various approaches to the environmental and economic analysis of sludge treatment and recycling systems (STRS). These investigations did not take the universal law of different capacities for environmental impact and STRS economics into account. The aim of this study was to analyze the scale effect of STRS with different technologies (i.e., incineration, aerobic composting, used in material (brick), anaerobic digestion) on the environment and economy. Moreover, the cost–benefit impact of introducing a carbon- trading mechanism into the STRS to achieve carbon neutrality was analyzed. After reducing carbon emissions through by-products of STRS, the carbon emission quota can be sold, which will generate income. The results show that the break-even scales for incineration, anaerobic composting, used in building material (brick), and anaerobic digestion are 54,899, 6707, 48,775, and 4425 t/y, respectively. The break-even scale of each system decreased after the introduction of the carbon trading system into the STRS. These findings could provide critical technical information for superior decision-making in sewage sludge recycling systems.

**Keywords:** sludge management; scale effect; ordinary least squares regression; greenhouse gas; carbon neutrality; cost benefit analysis

## 1. Introduction

Due to the acceleration of economic development and urbanization, the number of wastewater treatment plants has increased significantly, resulting in a rapid accumulation of sludge. The amount of sewage sludge generated with an 80% moisture content reached 39.04 Mt in 2019 in China [1,2]. This sludge requires immense space and causes greenhouse gas (GHG) emissions in landfills [1]. Leachate pollution from landfills contaminates groundwater with heavy metals, endocrine-disrupting compounds, and pharmaceutical and personal care products [3,4]. In addition, soil ecosystems can be affected by the accumulation of heavy metals. Inappropriate treatment causes secondary pollution from heavy metals, organic pollutants, pathogens, and dioxins, that severely threatens human health [5,6]. Consequently, sewage sludge disposal is considered a severe problem in wastewater plants and municipal waste management for local government.

Currently, many technologies are available for energy recovery, including mono-incineration, co-incineration, anaerobic digestion, pyrolysis, gasification, and supercritical wet oxidation [7]. The value-added by-products of resource recovery include biochar, adsorbents, fertilizers (phosphorus and nitrogen), and building materials (cement and brick) [8]. Technical requirements were proposed for sludge disposal according to the

Thirteenth National Urban Sewage Treatment and Reuse Facility Planning in December 2016 [9]. Sludge disposal facilities should be constructed in accordance with the principle of "combining centralized and decentralized treatment" to form scale effects and encourage resource recycling. Compared to the greenhouse gas (GHG) intensity of wastewater treatment in China (0.79 t/m$^3$), the optimal sludge recycling system could offset the GHG emissions, whereas the worst-case sludge recycling system would increase total GHG emissions by 149% [8]. Wei et al. [1] reported the GHG intensity of sewage sludge disposal, which considers the energy and resource recovery of four main technical routes in China: incineration, sanitary landfills, land utilization, and building materials. A previous study showed the environmental impact of six alternative scenarios with and without sewage sludge digestion combined with three end-of-life disposals [10]. The electricity consumption was negligible because of the reutilization of waste heat during the incineration and melting processes. Sewage sludge recycling via energy recovery or resource reuse can improve the neutrality of environmental impacts such as carbon emissions. We introduced a carbon trading mechanism for sludge treatment and recycling systems (STRSs) to contribute to the carbon neutrality of wastewater treatment systems in China [11].

The carbon trading system (CTS) incentivizes low-carbon practices and innovation strategies as tools for public policy reform. On 18 December 2017, the National Development and Reform Commission in China issued the "National Carbon Emission Trading Market (Power Generation Industry) Construction Plan", indicating that the national carbon emissions trading market has been formally established in China. The first carbon trading project was the Landfill Gas Clean Development Mechanism Project in Beijing [12]. Huang and Xu [13] conducted a bi-level multi-objective programming approach for coal and sewage sludge co-combustion, considering the maximization of economic benefits and sewage sludge utilization and the minimization of carbon emission. According to the formal establishment of the carbon market and carbon emission allowance by the government, carbon emissions can be assigned monetary value. Therefore, studying China's carbon trading is crucial to achieving emission reduction and sustainable development in wastewater treatment for STRSs.

Most studies of the environmental and economic performance of STRSs have focused only on technology selection and have not closely examined the effects of the implementation scale. Ignoring these effects made the comparison of different technology selections of SRTS according to environmental and economic performance uncertain. Chen et al. [14] analyzed the GHG potential mitigation of sludge recycling management considering the energy and resource recovery. It examined the economic efficiency of co-processing sludge with municipal waste and reported the limitations of the study, which did not consider the different capacities of incinerators. Other studies showed that energy consumption and operation cost are related to the implementation scale, and therefore the analyses were conducted on a large-scale sewage sludge recycling system (STRS) to avoid scale effects [9,15]. Luo et al. [2] presented the results of environmental and economic analysis for full-scale sludge pyrolysis systems and proved that a larger pyrolysis system for centralized sludge handling was more economically favorable. Kumar et al. [16] discussed the relationship between the payback period, return of investment, and plant capacity of biodiesel production generated by municipal sludge. In term of analyzing the cost and size of equipment, it was carried out by power function [16]. We analyze how different implementation scales affect the environmental emissions and economic costs of different STRSs to fill this assessment gap and meet practical requirements.

The aim of this study was to analyze the effects of the implementation scale that affects the environmental and economic performance of STRSs with different technological selections (i.e., incineration, aerobic composting, used in building material (bricks), and anaerobic digestion). First, we collected and analyzed the environmental and economic performance of the STRSs per unit, including unit carbon emissions, initial cost, operation cost, and revenue of the system. Next, we determined the effects of the implementation

scale and the environmental and economic performance of SRTS. In addition, the second objective was to identify the impact of the implementation scale after introducing the carbon trading mechanism (CTS) into the STRSs.

## 2. Materials and Methods

The ordinary least squares method (OLS) is commonly used for parameter estimation in functional relationships. For example, it was considered in the study of Mydland et al. [17] that investigated the economies of scale of Norwegian electricity distribution companies. Referring to Fragkias et al. [18], they examined the relationship between city size and $CO_2$ emissions for the United States metropolitan areas using OLS. This study used OLS to analyze the relationship between the implementation scale, environmental emissions, and economic cost of a sewage sludge recycling system (STRS) to support decision-making. The optimal system and break-even scale of each system were obtained with cost–benefit analysis considering the effect of the implementation scale. Following our emphasis on implementation scale effects, we hypothesized that environmental emissions and economic costs are closely related to the implementation scale and that their relationships could be measured according to a power function, as shown in Equation (1).

$$Y = ax^\beta \tag{1}$$

where Y measures environmental emissions and economic cost, a is a constant, x denotes the implementation scale, and β is the scaling exponent. This function acts as a baseline model to determine whether environmental emissions or economic costs were modeled with a power function relationship.

Economic data were obtained from market investigations and environmental impact assessment (EIA) reports of each sewage sludge recycling project in China. Owing to the lack of condition of sludge treatment and disposal plants in China [1], we assumed that the effect of the implementation scale in the SRTS is similar in different countries, and the operating situation of different countries only affects the size of scale effect. We aggregated the situation of sewage sludge recycling based on data from the Japan Sewage Works Association (JSWA), including the scale of facilities and consumption of energy and chemicals. Data of GHG emission factor for treatment, energy and chemical was from the Research Institute of Economy, Trade and Industry, Ministry of the Environment of Japan (MINE) and China products carbon footprint factors database. The quantity of data relevant to our subsequent validations was limited due to the lack of public data on SRTS.

### 2.1. Comparison Cases

Currently, aerobic composting, anaerobic digestion and biomass utilization, incineration and electricity generation, and use for building materials are the four common scenarios of sludge treatment and recycling technologies in China [1,9,19]. The overall scope of this study comprises the dewatering, treatment, and recycling processes involved in each system, as shown in Figure 1. Literature has reported that three types of anaerobic digestion, including mesophilic, thermophilic, and temperature-phased anaerobic digestion, show slight differences based on the results of LCA [20]. Liu et al. [21] reported that the endpoint environmental impact and economic cost differed slightly among four types of aerobic composting. In terms of environmental impact, fluid bed incinerators were less damaging than multiple-hearth incinerators [22]. Based on data from the incineration of the SRTS project (165), 123 projects used the fluid bed incinerator, which accounted for 74.5%. Accordingly, the effects of the different technologies in each scenario were ignored in this study.

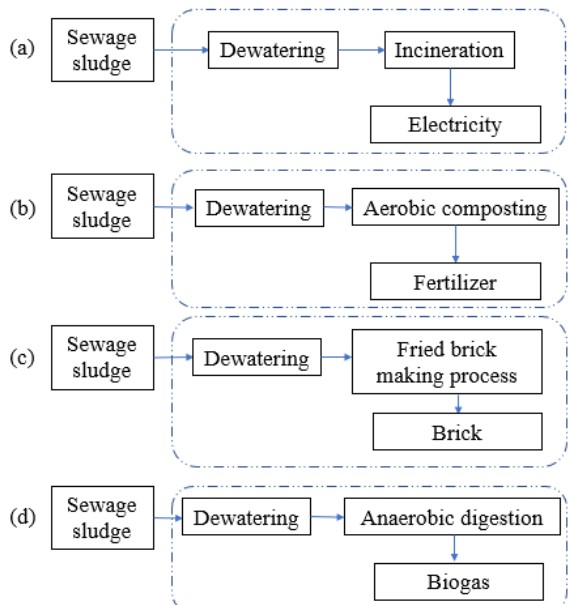

**Figure 1.** The boundaries of the target sewage sludge recycling system. (**a**) Incineration; (**b**) Aerobic composting; (**c**) Used in Building material (brick); (**d**) Anaerobic digestion.

The contribution of incineration increased from 17.3% in 2012 to 26.7% in 2019 in China [1]. Previous studies have reported that incineration plants exhibit economies of scale [23,24]. Sludge incineration eliminates viruses and pathogens and converts sludge organic matter into $CO_2$, which may be the reason that sludge incineration is recommended as a best practice in the sludge disposal area in the limited space available. Simultaneously, waste heat is consumed to generate electricity to reduce energy consumption over the incineration system. At present, the methods of incineration are decided into separate incineration and co-incineration with other wastes. This study focused on separate incineration to compare the effects of the implementation scale on different systems in the environment and economy [5,25,26].

Based on the results of the estimation on different scales for municipal waste, the economy of scale of aerobic composting was reported [27,28]. Aerobic composting is a 7-d retention process in which sludge is held at a temperature of at least 55 °C for a sufficient period to ensure complete composting. In general, sludge using aerobic composting and land application is one of the more economically viable options when conditions permit, compared to other sludge disposal methods. However, agricultural (aerobic) composting is currently considered one of the main methods of sludge recycling in Europe. In particular, the economic benefits of sludge as an organic fertilizer and urban greening are more obvious [29,30].

The third scenario is the use of sewage sludge in fired brick production as a raw material substitute. The study by Subrahmanya [31] showed that the economic performance and energy intensity are affected by the implementation scale of brick-making facilities. While heavy metals in the sludge can be completely stabilized during high-temperature roasting, the scenario solidifies heavy metals and makes full use of sludge. The limitation of the heavy metal content and leaching toxicity remains important. In this scenario, sewage sludge directly mixes with other raw materials, and it replaces part of the shale utilization, which reduces fly ash emissions and energy consumption during the breaking process [32–34].

Anaerobic digestion stabilizes sludge and avoids adverse effects on the environment during transportation and final disposal by degrading organic matter, thereby reducing the moisture content and quality of sewage sludge. The results of previous studies displayed that the economic performance of anaerobic digestion technologies for treating municipal solid waste were affected by economies of scale [35,36]. The sewage sludge is intermediate,

subjected to temperature anaerobic digestion, and produces biogas for sludge digestion and drying. When integrated with incineration and composting, digested sludge can be further recycled as electricity and fertilizer. In this scenario, we focused on obtaining recycling biogas from anaerobic digestion [7,25,37].

The main assumption of this study included the following four points.

(1) The GHG emissions of the construction phase were not examined as they did not exceed 5% of the total impact [19].

(2) Energy and by-products recovered in the SRTS were sold completely, regardless of market demand.

(3) The energy consumed by SRTS during the dewatering and treatment processes was derived from fossil fuels.

(4) The nitrogen content in the fertilizer generated by sewage sludge was 8%, which compared with the conventional fertilizer [38].

### 2.2. GHG Emission of Sewage Sludge Recycling System

In this study, we focus on GHG emissions to present the effects of the implementation scale and environmental emissions in each system. As the chemicals used in the treatment and recycling processes were far fewer than those used in the dewatering process, only the chemicals for the dewatering process were calculated. This study is based on the JSWA data, similar to the inventories of previous studies [10,19]. The GHG emissions calculated in this study, as described in Equation (2), include the chemical consumption in the dewatering process, the energy consumption of the system, and the discharge of sludge after treatment.

$$
\begin{aligned}
\text{GHG} &= \text{GHG}_{\text{treatment}} + \text{GHG}_{\text{energy}} + \text{GHG}_{\text{chemical}} \\
&= \text{EF}_{m,\text{treatment}} \times Q_{\text{treatment}} + \text{EF}_{i,\text{energy}} \times Q_{\text{energy}} + \text{EF}_{n,\text{chemicals}} \times Q_{\text{chemical}}
\end{aligned}
\tag{2}
$$

The following emission factors (EFs): $\text{EF}_{m,\text{treatment}}$, $\text{EF}_{i,\text{energy}}$, and $\text{EF}_{n,\text{chemical}}$ represent the discharge of sludge after the *m*-th kind of treatment, and pollution by consuming the *i*-th kind of energy and the *n*-th kind of chemicals, respectively. The $Q_{\text{treatment}}$, $Q_{\text{energy}}$, and $Q_{\text{chemicals}}$ represent the amount of dry solids (DS) of sewage sludge treatment, the amount of the *i*-th type of energy, and the amount of the nth type of chemicals, respectively. The calculated GHG emission factors are listed in Table 1.

**Table 1.** Greenhouse gas (GHG) emission factor (EF) calculated in STRS.

| Parameter | Unit | Value | Parameter | Unit | Value |
|---|---|---|---|---|---|
| Energy [2] | | | | | |
|    Heavy oil | $tCO_{2eq}/kL$ | 2.71 | LPG | $tCO_{2eq}/kL$ | 3 |
|    Coal oil | $tCO_{2eq}/kL$ | 2.49 | Disel | $tCO_{2eq}/kL$ | 2.58 |
|    Gasoline | $tCO_{2eq}/kL$ | 2.32 | Coal | $tCO_{2eq}/t$ | 2.33 |
|    Electricity | $tCO_{2eq}/kwh$ | 0.000433 | Natural gas | $tCO_{2eq}/10^3Nm^3$ | 2.62 |
| Chemicals [1,2] | | | | | |
|    Ferrous chloride | $tCO_{2eq}/t$ | 0.32 | Poly-ferrous sulfate | $tCO_{2eq}/t$ | 0.0308 |
|    Ca(OH)$_2$ | $tCO_{2eq}/t$ | 0.45 | CaO | $tCO_{2eq}/t$ | 0.75 |
|    PAM | $tCO_{2eq}/t$ | 6.5 | Poly-aluminum chloride | $tCO_{2eq}/t$ | 0.41 |
|    H$_2$O$_2$ | $tCO_{2eq}/t$ | 0.39 | | | |
| Sludge [2] | | | | | |
|    Incineration | $tCH_4/wet\text{-}t$ | 0.0000097 | Composting | $tCH_4/wet\text{-}t$ | 0.004 |
| | $tN_2O/wet\text{-}t$ | 0.0003 | | $tN_2O/wet\text{-}t$ | 0.0006042 |
| Production [3] | | | | | |
|    Electricity | $kgCO_{2eq}/kwh$ | 0.53 | Nitrogen Fertilizer | $tCO_{2eq}/t$ | 10.63 |
|    Clay Brick | $tCO_{2eq}/t$ | 0.2 | Biogas | $kgCO_{2eq}/t$ | 9.35 |

[1] Kainou [39]. [2] MINE [40]. [3] CAEP [41].

GHG emissions can be avoided if by-production replaces energy or substitution. The avoided GHG emissions (GHG$_{\text{avoided}}$) in large implementation scale conditions were lower

than the production emissions generated ($GHG_{product}$) by the original process of *k*-th product in every case, as shown in Equation (3).

$$GHG_{avoided} = GHG_{k,product} - GHG \tag{3}$$

### 2.3. Total Cost of Sewage Sludge Recycling System

In this study, we introduced CTS into the cost accounting of STRS. Carbon cost refers to the economic cost of purchasing or selling carbon emission rights in CTS. The cost accounting of the system is divided into four parts: the initial cost, operation cost, the cost of carbon emission, and by-product profit. The total cost per unit DS of each system included the initial cost of unit DS, operation cost of unit DS, and by-production profit of unit DS, as presented in Equation (4).

$$Cost_{total} = Cost_{initial} + Cost_{operation} + Cost_{by\text{-}product}$$

$$Cost_{operation} = Cost_{energy} + Cost_{chemical} + Cost_{carbon} \tag{4}$$

$$Cost_{carbon} = GHG \times P_{carbon}$$

where $Cost_{operation}$ represents the operation cost per unit of DS including the costs of energy consumption, chemical consumption, and carbon emissions. $P_{carbon}$ is the carbon price, which is the average market price in emission exchange. An exchange rate of 1 USD = 6.7 CNY was used, and all financial factors in Table 2 were transformed to USD.

**Table 2.** Financial parameters required to calculate STRS operation cost.

| Parameter | Unit | Value | Parameter | Unit | Value |
|---|---|---|---|---|---|
| Energy | | | | | |
| Heavy oil | USD/L | 0.71 | LPG | USD/m$^3$ | 1.87 |
| Coal oil | USD/L | 0.42 | Disel | USD/L | 1.03 |
| Gasoline | USD/L | 1.05 | Coal | USD/t | 253.23 |
| Electricity | USD/kwh | 0.095 | Natural gas | USD/m$^3$ | 0.39 |
| Chemicals | | | | | |
| Ferrous chloride | USD/t | 74.63 | Poly-ferrous sulfate | USD/t | 134.33 |
| Ca(OH)$_2$ | USD/t | 74.63 | CaO | USD/t | 67.16 |
| Polymer flocculant (PAM) | USD/t | 895.52 | Poly-aluminum chloride | USD/t | 179.10 |
| H$_2$O$_2$ | USD/t | 111.94 | CaCO$_3$ | USD/t | 59.70 |
| NaOH | USD/t | 223.88 | | | |
| By-production | | | | | |
| Clay Brick | USD/piece | 0.075 | Fertilizer | USD/t | 344.78 |
| Electricity | USD/kwh | 0.097 | | | |

With the introduction of the carbon emission quota (CEQ) into SRTS, the cost of the effective difference in carbon emissions should be considered in the cost accounting of the system, as presented in Equation (5). For example, GHG emissions generated during incineration can replace the carbon credit for electricity substitution [14,42].

$$Cost_{total}' = Cost_{initial} + Cost_{operation} + Cost_{by\text{-}product} + Cost_{CEQ}$$

$$Cost_{CEQ} = GHG_{avoided} \times P_{carbon} \tag{5}$$

where $Cost_{total}'$ is the total cost per unit of DS with the CEQ, and $Cost_{CEQ}$ is the cost of carbon credit via by-product substitution.

### 3. Results

### 3.1. Impact of Scale on GHG Emission of System

The unit GHG emission was related to the implementation scale in each system, and first decreased rapidly and then gradually stabilized as the scale increased, as shown in Figure S1. When the implementation scale was more than 209,178 t, the unit GHG emission

of incineration stabilized at 0.14 $tCO_2$/t-DS. The unit GHG emission of aerobic composting remained steady at approximately 0.18 $tCO_2$/t-DS when the implementation scale increased over 140,000 t. For the system used in building material (brick), the unit GHG emission fell to a low point around 0.01 $tCO_2$/t-DS over 7676 t of scale. Above the implementation scale of 40,000 t, the unit GHG emission reached approximately 0.08 $tCO_2$/t-DS. The energy consumption in the system was the main reason for determining the unit GHG emissions.

The avoided GHG emissions were calculated after introducing the CEQ to the system. Figure 2 shows that if the units of avoided GHG emissions had a negative value, the unit GHG emission of the system was greater than the GHG emission of production generated by the original process [43]. In contrast, if the avoided GHG emissions had a positive value, the system offset part of the carbon burden of production generated. The minimization scale of the balance of GHG emissions was calculated based on the scaling effect of environmental emissions. As shown in Figure 2, the minimization scales of incineration, aerobic composting, used in building material, and anaerobic digestion were 31,946, 19, 33, and 82 t-DS/y, respectively. Surprisingly, the minimization scale of incineration was much larger than that of the other systems. The increase in renewable electricity generation and expansion of cross-regional grid construction were the reasons for the decrease in the GHG emission intensity of electricity generation [44].

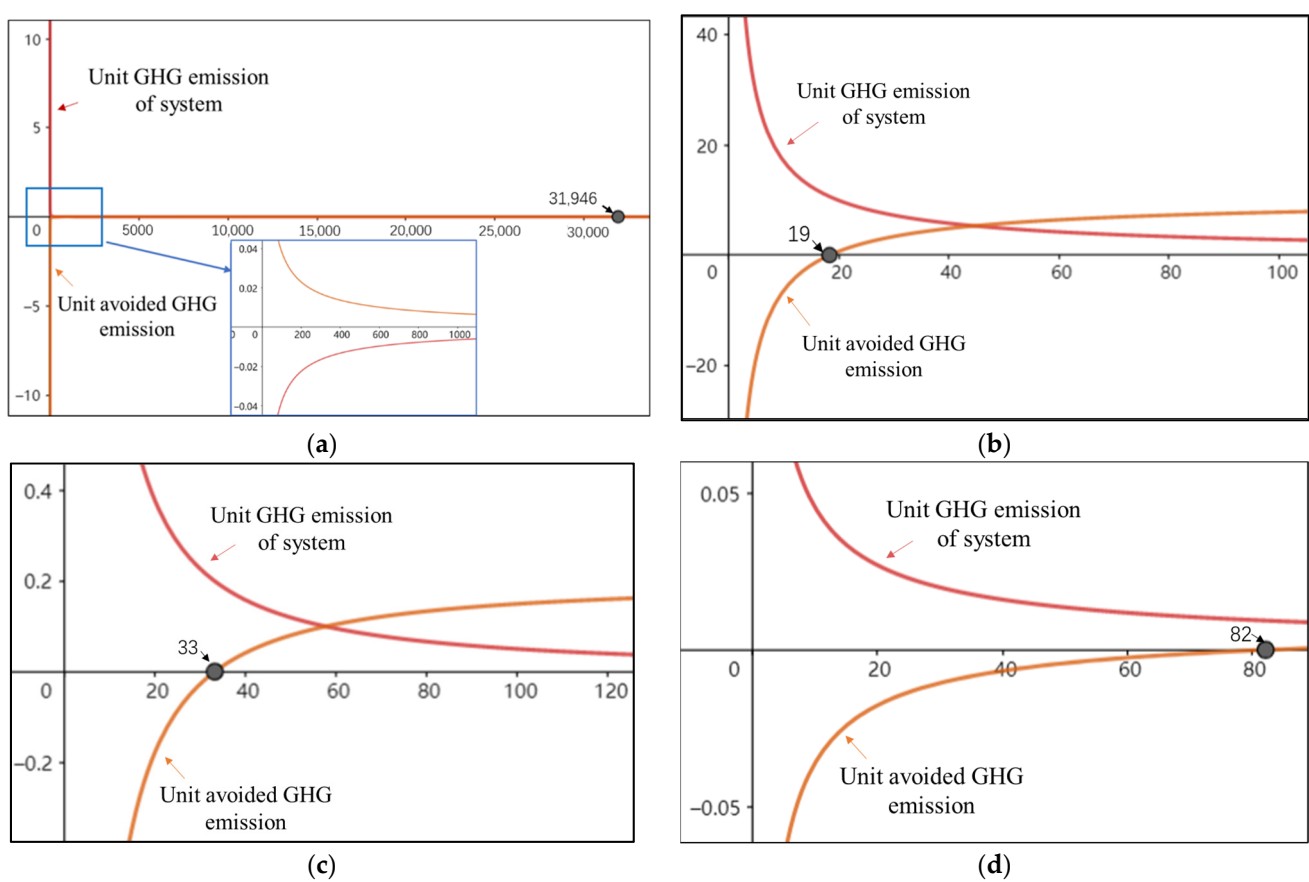

**Figure 2.** The unit avoided GHG emission of different systems. (**a**) Incineration; (**b**) Aerobic composting; (**c**) Used in Building material (brick); (**d**) Anaerobic digestion. Horizontal axis: the implementation scale of SRTS (t-DS/y). Vertical axis: the unit GHG emission (kg $CO_2$ eq/t by-products).

### 3.2. Impact of Scale on Cost and Benefit of System

### 3.2.1. The Unit Initial Cost

Initial cost is a critical factor directing decision-making for investors. Investors determine whether projects are viable through the initial cost analysis and finance projections. In this study, we assumed the lifetime of the sewage sludge recycling facility to be 30 years [2].

The unit initial cost decreased with an increase in implementation scale. The scaling exponent of unit initial cost is about 0.438 to 0.606 as shown in Figure S2. The results in Figure S2 illustrate the significant difference between the initial cost of use in building material and the other three systems; the latter had approximately seven times greater cost than the former. A comparison of the proportion of initial cost in the annual cost accounting revealed that anaerobic digestion had an initial cost of approximately 25% of the whole which was the highest proportion, and the lowest one was used in building material (approximately 8%) under the implementation scale of 10 kt DS.

### 3.2.2. The Unit Cost of Energy Consumption

In the entire system, energy consumption was due to the utilization of electricity, coal, A heavy oil, liquefied petroleum gas (LPG), kerosene, diesel, gasoline, and natural gas. The price of energy was collected from published data from the government and the China Petroleum & Chemical Industry Association. The unit cost of energy consumption was obtained by total energy consumption of the system collected by EIA dividing the annual scale. The implementation scale had a negative impact on the unit cost of energy consumption, as shown in Figure S3. It provided 63.7%, 24%, 1.9%, and 37.9% of the unit operation cost for incineration, aerobic composting, use in building material, and anaerobic digestion, which treated 10 kt-DS per year, respectively.

### 3.2.3. The Unit Cost of Chemical

Further statistics revealed that the unit cost of chemical consumption followed the scale effect, which decreased as the implementation scale increased. During the dewatering process in each system, the critical part of the cost of chemicals is PAM. The reduction rate of 200 kt to 1200 kt and 2 Mt to 3 Mt decreased from 76% to 27% and gradually stabilized, as shown in Figure 3.

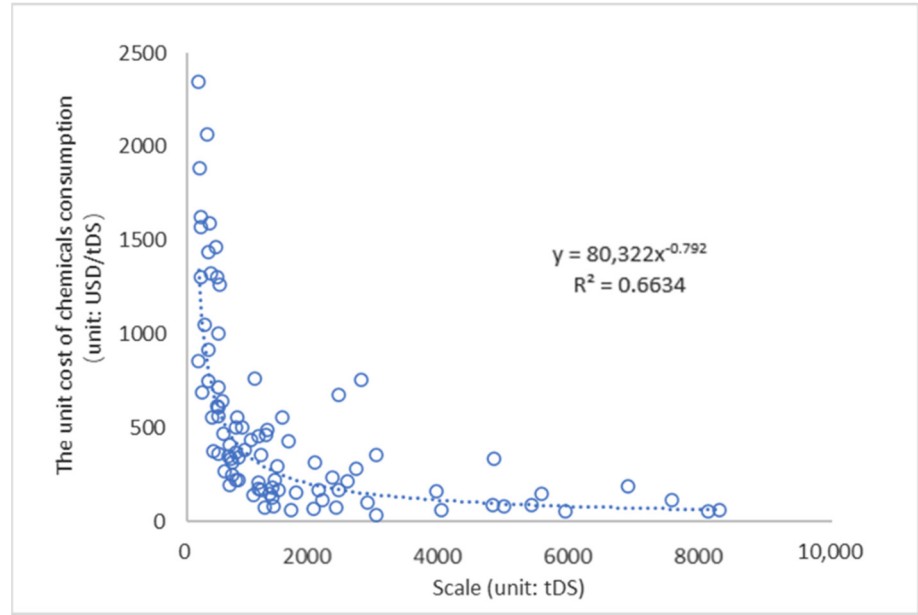

**Figure 3.** The relationship of the unit cost of chemicals consumption and implementation scale. Horizontal axis: the implementation of scale of SRTS projects (t-DS). Vertical axis: the unit cost of chemicals consumption (USD/t-DS).

### 3.2.4. The Unit Cost of Carbon Emission

The regular pattern follows a power function similar to the GHG emissions presented in Section 3.1. The cost of carbon emissions refers to the direct carbon emissions from the system and does not include the carbon emissions tax. The cost accounting of the system

appeared to be unaffected by the cost of carbon emissions, even when the carbon trading mechanism was introduced.

### 3.2.5. Revenue of By-Products

A positive correlation was found between the implementation scale and unit revenue of each system. The data in Figure S4d can be compared with the data in Figure S4a–c, which shows that anaerobic digestion has a clear trend of increasing at the beginning of the increase in the implementation scale. The unit revenues of the by-products of anaerobic digestion and aerobic composting are higher than those of the other two systems. The productivity growth rate steadily increased with the increase in implementation scale. This phenomenon was attributed to the low production efficiency caused by the low calorific value and C/N ratio of the sewage sludge [11,45,46].

### 3.2.6. The Total Cost

The total cost in this study was the sum of the cost accounting and revenue of each system. Dividing the total annual cost by the annual scale yields the total unit cost. The first requirement for sewage sludge recycling is the initial cost minimization when investors or policy-makers decide to invest in the project. Figure 4 shows that while the decision-maker does not consider the revenue of the system, the optimal technology is used in building material (brick) with the lowest unit initial cost and unit operation cost. The results of previous studies also show that the building material system was the optimal selection based on the lowest economic cost [9].

Data from Figure 4a can be compared with the data in Figure 4b, which show the contribution of the initial cost and operation cost. In the system of incineration to treat 1 Gt dry solid of sludge, the operation cost covered 80%, of which 85.3% was from the cost of energy consumption. The operational cost was 31% of the total cost of aerobic composting, where the cost of energy and chemical consumption was approximately equal. The operation cost was 17% of the total cost in the system of use in building material, which represents the cost of the predominantly used chemicals. The costs of energy and chemical consumption inside the operation cost of anaerobic digestion were 54% and 18%, respectively.

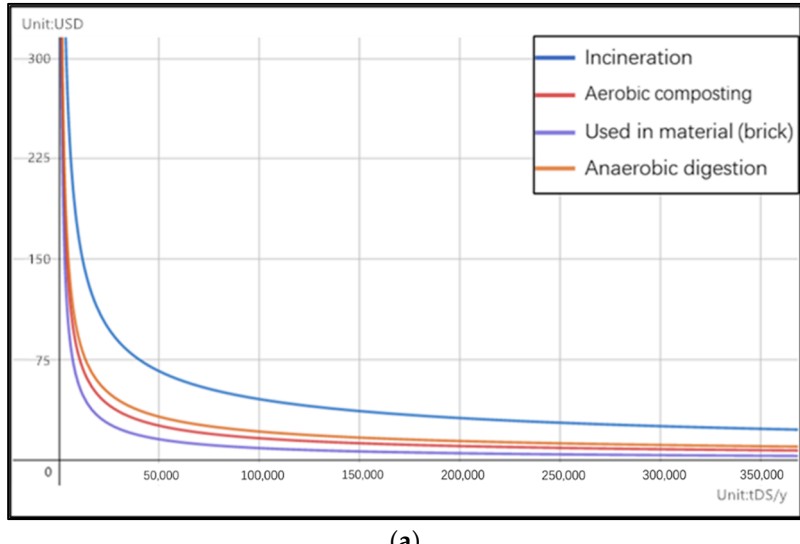

(**a**)

**Figure 4.** *Cont.*

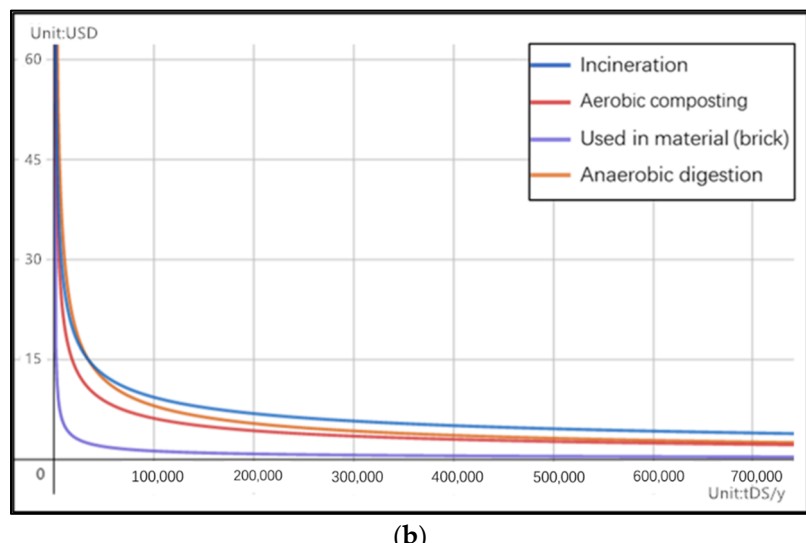

(**b**)

**Figure 4.** The unit cost accounting of different system. (**a**) The operation cost of four systems included the cost of energy and chemical consumption, and carbon emission cost; (**b**) Initial cost. Horizontal axis: the implementation scale of SRTS (t-DS/y). Vertical axis: the unit operation cost and unit initial cost of four systems (USD), respectively.

The revenue of anaerobic digestion has a clear increasing trend corresponding to the beginning of the implementation scale increases, as shown in Figure 5. The unit revenues of the by-products of anaerobic digestion and aerobic composting are obviously higher than those of the other two systems. The optimal system based on the unit revenue of by-products is anaerobic digestion, which has an implementation scale of less than 297,255 t −DS, and the favorable system is aerobic composting.

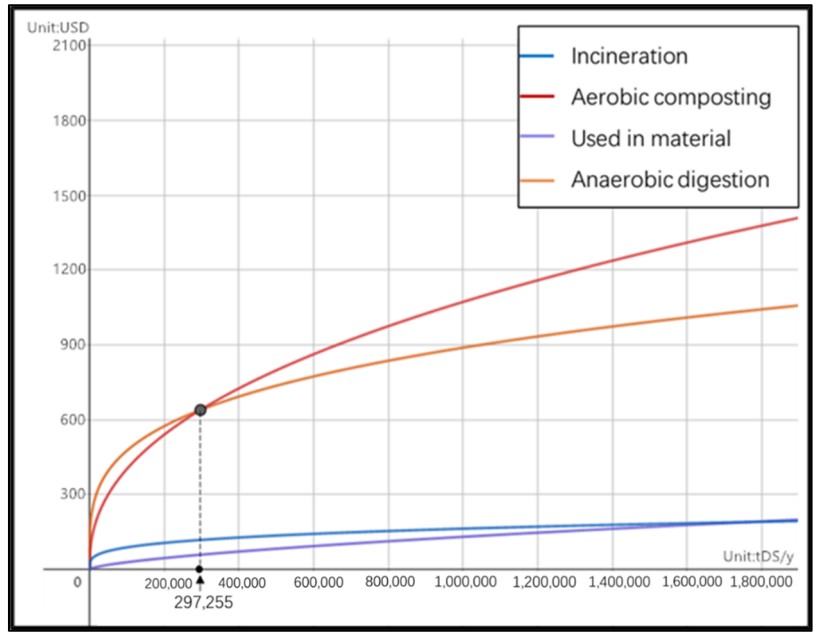

**Figure 5.** The unit cost accounting of different system with the revenue of by-product. Horizontal axis: the implementation of scale (t-DS/y). Vertical axis: the revenue from by-products of SRTS (USD).

### 3.3. The Break-Even Implantation of Scale

The curve in Figure 6 shows the total cost of each scenario. The break-even scale refers to revenue equal to the cost of the system as the zero point of the curve. Therefore, the project had financial value, and the scale implementation exceeded the break-even scale

without government subsidies. The minimum break-even scale was anaerobic digestion (4425 t-DS/y), followed by the break-even scale of aerobic composting (6707 t-DS/y), used in building material (48,775 t-DS/y), and incineration (54,899 t-DS/y). In summary, the results showed that the sewage sludge recycling system had negative economic performance when the implementation scale was lower than 4425 t-DS/y. The optimal system was anaerobic digestion, with an implementation scale between 4425 and 285,345 t-DS/y. With successive increases in the implementation scale exceeding 285,345 t-DS/y, the aerobic composting system was the optimal system.

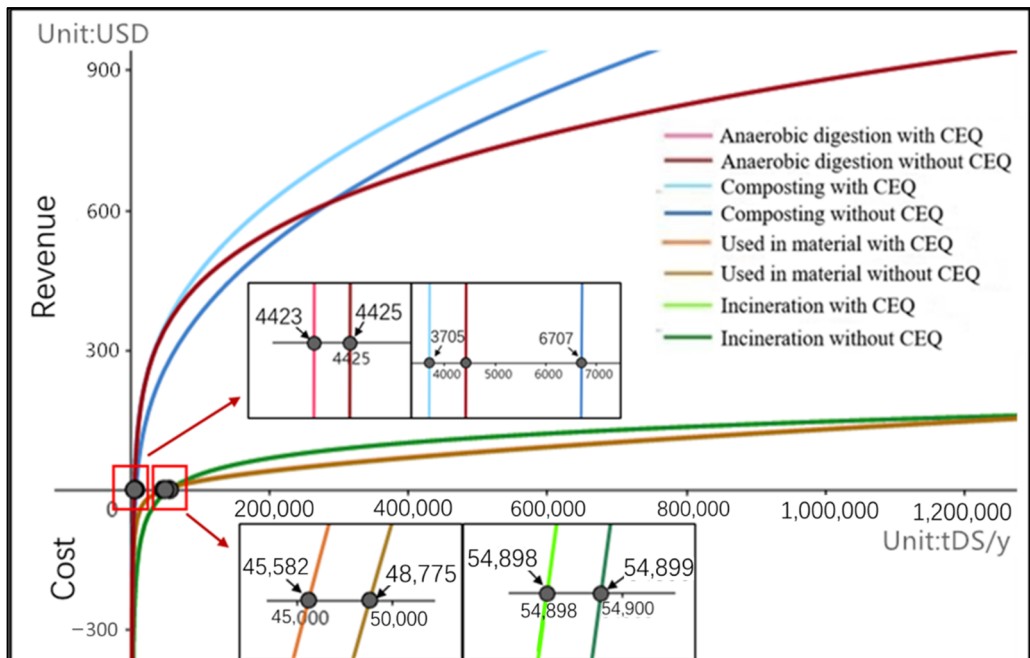

**Figure 6.** The break-even scale of each system considering carbon emission quota (CEQ). Horizontal axis: the implementation scale of STRS (t-DS/y). Vertical axis indicated the unit total cost (USD).

Based on the CEQ of production and China's carbon trading mechanism, the avoided GHG emissions can obtain revenue via carbon emission trading. Subsequently, the impact of introducing the carbon trading mechanism to the break-even scale was analyzed. Comparing the break-even scale with and without the CEQ, there was a significant decrease in the break-even scale of aerobic composting. Nitrogen fertilizer had the highest carbon emissions in the by-products of all systems. Therefore, the impact of the break-even scale with the CEQ was significantly higher than that in other systems.

*3.4. Sensitivity Analysis*

With the introduction of the carbon trading mechanism, sewage sludge recycling facility investors can earn revenue by selling CEQ, which in turn decreases the total cost of the system. The break-even scale can be affected by the cost of carbon emissions and revenue of avoided carbon emissions. The data of carbon trading market in each pilot city shown, violent fluctuation in the carbon price was caused by the policy related to car-bon emission allowance and is due to the immature market of China's emission trading scheme with an obvious policy-oriented [12]. Therefore, we conducted a sensitivity analysis for the cost of carbon emissions and break-even scales using different carbon prices. As shown in Figure 7, the rate of unit carbon emissions in the total cost had an apparent difference between incineration and aerobic composting and the other two systems. The rate of unit carbon emissions in total cost increased with an increasing carbon price; among them, the system with the highest ratio was aerobic composting (2.39%) and the lowest was used in building material (0.11%).

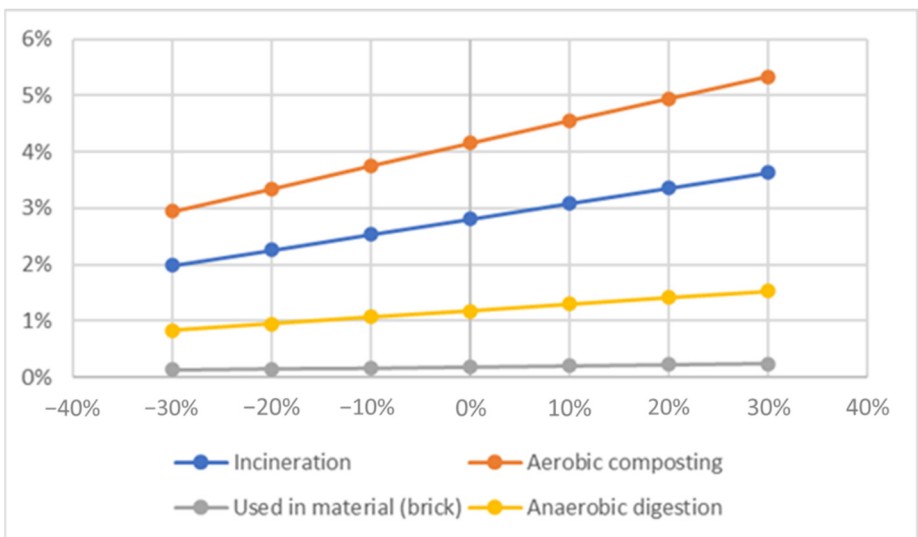

**Figure 7.** Relationship between the rate od unit carbon cost in total cost and carbon price. Horizontal axis: the change in carbon price (%). Vertical axis: the rate of the unit cost of carbon emission in unit total cost (%).

With the increase in the carbon price, the break-even scale without the CEQ increased slightly. Further analysis showed that the break-even scale with CEQ of aerobic composting and use in building material decreased owing to the profit from the avoided carbon mission. Significantly, Figure 8 shows that the break-even scale with CEQ of incineration and anaerobic digestion increased, which is approximate to the break-even scale without CEQ. When the break-even scale with CEQ decreased as the carbon price increased, the unit cost of carbon emissions was larger than the profit from the avoided carbon emission of the product. Therefore, sewage sludge recycling technology is recommended to replace high-carbon-emission products.

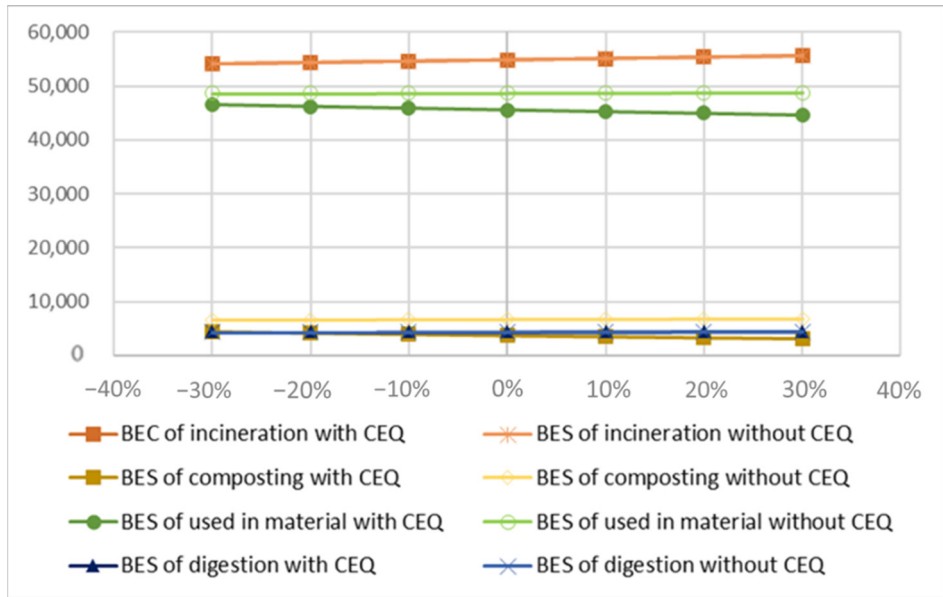

**Figure 8.** The sensitivity analysis of the break-even scale and carbon price. Horizontal axis: the change in carbon price (%). Vertical axis: the break-even scale of SRTS in different carbon price (t-DS/y).

## 4. Conclusions

Due to rapid urbanization in China, sewage sludge production is expected to grow rapidly. The amount of sewage sludge generated is closely related to the scale of the city [1]. Meanwhile, the policy of the "zero waste" city and the high amount of sewage sludge generated provided an enormous opportunity for waste valorization. Sewage sludge recycling systems can substitute products and generate income by reducing GHG emissions [19]. The findings of this study make several contributions to the current literature. One significance of the study is that it provides new evidence that the implementation of scale affects the results of environmental and economic assessments of technology selection for sewage sludge recycling systems [9]. A quantitative comparative evaluation was conducted on the GHG emissions and the unit total cost of each system.

The unit cost of each part decreased with the increase in the implementation scale, while the unit revenue of by-products increased with the implementation scale [16]. Therefore, considering the revenue of by-products, there is no economic value in which the unit total cost is negative when the actual scale is smaller than the break-even scale of each system for the investors without government subsidies. If the policy-oriented project ignores the implementation scale, it would reduce the financial burden on the local government. Unit carbon emissions decreased as the implementation scale increased. When introducing the carbon trading mechanism, it is advantageous that it minimizes the break-even scale under the income generated from the avoided carbon emission allowance, to expand the applicability of sewage sludge recycling [12].

The insights gained from this study improve the accuracy of economic and environmental evaluations of project performances. For investors, the decisive factor in starting a new project was the total initial cost of different technology selections. The optimal technology for the sewage sludge recycling project was used in building material (brick) [47], which merely considered the initial cost of the project and ignored the impact of the implementation scale. Considering the implementation scale, aerobic composting and anaerobic digestion were the optimal technologies on different implementation scales, respectively. Hence, the technology selection and implementation scale were critical elements of the sewage sludge recycling-system strategy for decision-makers based on the scale of the city [1,9].

In conclusion, this study successfully quantified the impact of the implementation scale on the environmental and economic performance of different sewage sludge recycling systems. The main findings of this study are as follows:

(1)  The small implementation scale of the GHG emission balance was determined by considering the substitution of energy and resources. The small implementation scales of incineration, aerobic composting, use in building materials (bricks), and anaerobic digestion were 31,946, 19, 33, and 82 t-DS/y, respectively.

(2)  When considering the subsidy and substitution of energy and resources, the break-even scales of incineration, aerobic composting, use in building materials (bricks), and anaerobic digestion were 54,899, 6707, 48,775, and 4425 t-DS/y, respectively. The break-even scale was reduced by introducing a carbon trading system into the sewage sludge recycling system.

(3)  The optimal technology for different implementation scales was determined. Aerobic composting was a prior technology used when the implementation scale was larger than 285,345 t-DS/y. Anaerobic digestion was prioritized when the implementation scale was between 4425 t-DS/y and 285,345 t-DS/y.

A limitation of this study is the current lack of conditions of sewage sludge recycling projects in China [19]. The lack of data resulted in a less precise scale exponent and break-even scale. We strongly suggest that decision-makers should pay attention to intelligence statistics and forceful administration. In any case, this study presents a preliminary analysis of suitable technical routes for sewage sludge recycling systems considering the scale effect. In the future, the results of the implementation scale can be used by policymakers, designers, and investors in the preplanning and project renovation stages. At the local government

level, sustainable sewage sludge management should be designed according to the specific needs of each city. Future research should investigate the effect of regional characteristics on market demand for substitutions, operation costs, and subsidies for sewage sludge disposal to contribute to a complete sewage sludge management system.

**Supplementary Materials:** The following supporting information can be downloaded at: https://www.mdpi.com/article/10.3390/su14148684/s1. Figure S1: The relationship of unit GHG emission of different system and implementation scale; Figure S2: The relationship of unit initial cost of different system and the implementation scale; Figure S3: The relationship of the unit cost of energy consumption of different system and the implementation scale; Figure S4: The relationship of the unit revenue of different system and implementation of scale.

**Author Contributions:** Conceptualization, J.Z. and T.M.; methodology, J.Z., T.M. and Z.L.; software, J.Z.; validation, J.Z., T.M. and Z.L.; investigation, T.Z. and J.Z.; resources, T.M.; writing—original draft preparation, J.Z.; writing—review and editing, J.Z., Z.L. and T.M.; supervision, T.M. and Z.L.; funding acquisition, T.M. All authors have read and agreed to the published version of the manuscript.

**Funding:** This research received no external funding.

**Institutional Review Board Statement:** Not applicable.

**Informed Consent Statement:** Not applicable.

**Data Availability Statement:** Not applicable.

**Conflicts of Interest:** The authors declare no conflict of interest.

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
