# Peer review of "Environmental and Economic Implication of Implementation Scale of Sewage Sludge Recycling Systems Considering Carbon Trading Price"

_sustainability, doi:10.3390/su14148684_

Round 1
Reviewer 1 Report
The topic of the paper under the title “Environmental and Economic Implication of Implementation Scale of Sewage Sludge Recycling Systems Considering Carbon Trading Price” is current and within the scope of the Sustainability journal. However, the manuscript is prepared in a chaotic manner and some issues require clarification and improvement. Therefore, I recommend major revisions before the publication of the manuscript.
1. The keywords should be different from the words in the title. Please change or delete the following keywords: "Sludge recycling", "Carbon trading".
2. The Introduction needs to be improved and expanded. Please refer to more studies on the recycling of sewage sludge. In the current version of the manuscript, the literature review is rather poor. In lines 32-33, please add the country which the given data refers to. It is likely China, but this is not apparent from the content of the manuscript. In lines 65, 69 and 72, I suggest inserting the literature reference (number in square brackets) instead of the year.
3. In section 2, please justify why the effects of the different technologies in each scenario have been ignored (lines 112-113). This is relevant to the research results obtained and requires explanation. Simply stating that certain issues have been ignored is not sufficient. Please also avoid using lumped references (line 132). The sentence in lines 149-151 needs to be confirmed by literature data.
4. Are the authors able to provide the data from which the values shown in Table 1 were calculated?
5. The authors provide costs and unit prices in yuan. This currency is not widely known. For this reason, please additionally provide values in euros or dollars. This applies to the text as well as to tables and figures. For example, the authors can add an additional axis on which the values are presented in another currency.
6. Lines 215-219 – Figure S1 shows unit GHG emissions, not unit initial costs. Please correct the reference to the Supplementary Materials. Where did the value of 0.408 come from? There is also a problem with other references to Supplementary Materials. Figure S2 includes unit initial cost, not unit cost of energy consumption (lines 229-230). These data are shown in Figure S3. Can a trend line be drawn from just 3 points (Fig. S3c)? The same is true of the reference to Figure S4 (line 249). Reference to one of the figures is missing.
7. Axis descriptions are missing in the figures. Please complete the descriptions of the vertical and horizontal axes in all figures. In addition, Figure 2a is illegible. Please also check the numbering of the figures. There are two figures 5 and two figures 6 in the manuscript.
8. There are two identical sections in the manuscript differing only in the title (sections 4 and 5). Please delete one of them.
9. The text contains many editorial errors. For example, there should not be a full stop before square brackets (e.g. in lines 38, 53,56, etc.). Please put "STRS" in brackets (line 89). There is a typo in line 56 ("caron"). All equations should have a reference in the text, including equation (1). There is an unnecessary letter "T" in line 96. These are just some examples. Please check and correct the entire text of the manuscript carefully.
Reviewer 2 Report
This manuscript investigates the scale effect of sludge treatment and recycling systems (STRS) with different technologies (i.e., incineration, aerobic composting, used in material (brick), anaerobic digestion) on the environment and economy. Also, it studies the cost-benefit impact of introducing a carbon trading mechanism into the STRS to achieve carbon neutrality.
1) What is the research hypothesis?
2) Sections 4 and 5 must be merged.
3) Why did you use the ordinary least squares regression (OLS) for analysis?
Reviewer 3 Report
Zhang et al., studied about the “Environmental and Economic Implication of Implementation Scale of Sewage Sludge Recycling Systems Considering Car- -bon Trading Price: parameter assessment and cost exploration” It is interesting but the paper can be accepted for publication only after minor revision.
1. Describe more on the environmental issues in the introduction
2. Kindly improve on the discussion. What is the significance of the results of the work? Include more relevant literatures
3. Future scope and perspectives are very general
Round 2
Reviewer 1 Report
Thank you for considering my comments. In my opinion, the paper can be published.